# Beat-Notes Acquisition of Laser Heterodyne Interference Signal for Space Gravitational Wave Detection

**DOI:** 10.3390/s23063124

**Published:** 2023-03-15

**Authors:** Zhenpeng Wang, Tao Yu, Yanlin Sui, Zhi Wang

**Affiliations:** 1Changchun Institute of Optics, Fine Mechanics and Physics, Chinese Academy of Sciences, Changchun 130033, China; 2University of Chinese Academy of Sciences, Beijing 100049, China; 3School of Electronic Information Engineering, Changchun University of Science and Technology, Changchun 130022, China; 4School of Fundamental Physics and Mathematical Sciences, Hangzhou Institute for Advanced Study, UCAS, Hangzhou 310024, China

**Keywords:** frequency acquisition, space gravitational wave detection, laser heterodyne interference, Doppler shift

## Abstract

In space gravitational wave detection missions, the laser heterodyne interference signal (LHI signal) has a high-dynamic characteristic due to the Doppler shift. Therefore, the three beat-notes frequencies of the LHI signal are changeable and unknown. This may further lead to the unlocking of the digital phase-locked loop (DPLL). Traditionally, fast Fourier transform (FFT) has been used as a method for frequency estimation. However, the estimation accuracy cannot meet the requirement of space missions because of the limited spectrum resolution. In order to improve the multi-frequency estimation accuracy, a method based on center of gravity (COG) is proposed. The method improves the estimation accuracy by using the amplitude of the peak points and the neighboring points of the discrete spectrum. For different windows that may be used for signal sampling, a general expression for multi-frequency correction of the windowed signal is derived. Meanwhile, a method based on error integration to reduce the acquisition error is proposed, which solves the problem of acquisition accuracy degradation caused by communication codes. The experimental results show that the multi-frequency acquisition method is able to accurately acquire the three beat-notes of the LHI signal and meet the requirement of space missions.

## 1. Introduction

Gravitational wave detection in space is based on laser heterodyne interferometry [1]. By measuring the phase change of the LHI signal, the distance change information between different satellite test masses can be calculated [2], so as to achieve gravitational wave signal inversion. However, because the displacement change caused by the gravitational wave signal is very weak, many other functions need to be implemented to ensure the measurement accuracy, for example, inter-satellite ranging and communication, and pilot tone jitter correction [3]. Therefore, the components of the LHI signal are very complicated, including the main carrier beat-note (carrier beat-note), two clock sideband beat-notes (two side beat-notes or side beat-notes), communication codes, and various noises [4]. The distance change information can be obtained by measuring the phase changes of three beat-notes.

The phasemeter is the payload for high-accuracy phase measurement [5]. The phasemeter applies the principle of the digital phase-locked loop to measure the phase change of three beat-notes. However, due to the relative speed between spacecraft, when the laser travels from one spacecraft to another, the Doppler shift [6,7] will shift the frequencies of three beat-notes. Take the LISA as an example [1,2,7]: the frequency range can reach 9 MHz [3], resulting in unknown frequencies of three beat-notes. This may lead to the locking time of the digital phase-locked loop being too long, or even out of lock [8], which will lead to phase information errors. Therefore, before measuring the phase, it is necessary to accurately calculate the frequency of the three beat-notes.

The design of the LISA frequency acquisition algorithm for the carrier beat-note was based on FFT [3]. However, due to the picket fence effect and spectrum leakage of the discrete spectrum [9,10], the frequency error is large [11], and can not meet the requirements of high accuracy. Taiji Plan is a space gravitational wave detection plan proposed by China Academy of Sciences. In the Taiji plan, the current set frequency acquisition accuracy is not less than 30 Hz, and this index may be further improved in the future.

In the fields of harmonic detection [12] and power systems [13,14,15], the interpolated discrete Fourier method has been proposed to analyze exponential signals and estimate frequency [16]. Additionally, the accuracy of frequency estimation is further improved by different interpolation methods [17,18,19,20]. However, these methods are difficult to realize and take up a lot of resources. In addition, based on discrete wavelet packet transformation [21], all-phase [22], decomposition filtering-based dual-window correction algorithms [23], autocorrelation [24], and neural network methods [25], the frequency estimation can be realized. The influence of white noise on frequency estimation has also been analyzed [26,27]. Notably, the COG method is often use for position measurements [28,29]. The real position of the discrete spectrum peak can be obtained by the amplitude ratio of the discrete spectrum. Therefore, applying the COG method after FFT can estimate frequencies. Compared with other methods, the COG method has the characteristics of easy implementation and simple operation, but the estimation accuracy is easily affected by communication codes or noises.

We therefore propose a high-accuracy multi-frequency acquisition method based on the COG method. The method improves the estimation accuracy by using the amplitude of the peak points and the neighboring points of the discrete spectrum. For different windows that may be used for signal sampling, a general expression for multi-frequency correction of the windowed signal is derived. Meanwhile, a method based on error integration to reduce the acquisition error is proposed, which solves the problem of acquisition accuracy degradation caused by communication codes. The proposed method is implemented in VHDL based on the Field Programmable Gate Array (FPGA) platform and experimentally verified.

The structure of the paper is as follows. Section 2 describes the composition of the LHI signal. Section 3 derives a general expression for multi-frequency correction. The error integration method is illustrated in Section 4. Section 5 shows the simulation results. Section 6 describes the implementation of the method on an FPGA platform. Section 7 describes the experimental facilities and test results. Section 8 gives the conclusions.

## 2. Composition of the LHI Signal

In the LISA, the initial setting of the frequency of carrier beat-note is 11 MHz and the frequencies of side beat-notes are 11 MHz ± 1 MHz. The relative velocities of the two spacecraft can reach 10 ms^−1^. When the laser wavelength is 1064 nm, the maximum Doppler shift is defined as:(1)$$\Delta f=f\times \left(\frac{\sqrt{c+\Delta v}}{\sqrt{c-\Delta v}}-1\right)\approx \frac{\Delta v\cdot f}{c}=\frac{\Delta v}{c\lambda }=9\mathrm{MHz}$$
where Δf is the maximum Doppler shift, f is the laser frequency, c is the speed of light, Δv is the relative velocity and λ is the laser wavelength.

Three frequencies of beat-notes are unknown due to Doppler shift, as shown in Figure 1. The frequency of carrier beat-note ranges from 2 MHz to 20 MHz, and the range of frequency variation of the LHI signal is 1 MHz to 21 MHz.

Reference [4] derived the single quadrant voltage formula of the LHI signal output through a Trans-Impedance Amplifier (TIA):(2)$$\begin{array}{c} V=J_{0}^{2}\left(m_{sb}\right)\sin\left[2\pi \left(f_{m}+f_{D}\right)t+\varphi_{m}+m_{prn}c_{n}+n\left(t\right)\right]\\ +J_{1}^{2}\left(m_{sb}\right)\left\{\sin\left[2\pi \left(f_{u}+f_{D}\right)t+\varphi_{u}+n\left(t\right)\right]+\sin\left[2\pi \left(f_{l}+f_{D}\right)t+\varphi_{l}+n\left(t\right)\right]\right\} \end{array}$$
where $J_{k}\left(m\right)$ is the first kind of k-order Bessel function, and $m_{sb}\;\approx \;0.45$ is the phase modulation index. $f_{m},f_{u},f_{l}$ are the frequencies of three beat-notes, respectively, $f_{D}$ is the Doppler shift, and $\varphi_{m},\varphi_{u},\varphi_{l}$ are the phases of three beat-notes, respectively; $m_{prn}=0.1^{}\;\mathrm{rad}$ is the communication modulation index, and $c_{n}$ is communication codes composed of binary [–1, 1] sequences; $n\left(t\right)$ stands for noise. The amplitude ratio of three beat-notes is about 18:1:1. The logarithmic spectrum of the LHI signal is shown in Figure 2.

## 3. Multi-Frequency Correction

### 3.1. Limitations of the Traditional Method

Traditional frequency estimation algorithms are generally based on the FFT, which converts a signal in the time domain into a discrete spectral sequence. The maximum value of the spectral sequence corresponds to the frequency of the signal. The expression is defined as:(3)$$X\left(k\right)=\sum_{n=0}^{N-1}x_{n}e^{-j\frac{2\pi nk}{N}},\;0\;\leq \;k\;\leq \;N-1$$
(4)$$f=\frac{\mathrm{find}\;\max\left[k,abs\left(X\left(k\right)\right)\right]}{N}f_{s}$$
where N is the number of sampling points, $x_{n}$ is the sampling signal, $X\left(k\right)$ is the spectrum sequence, $\mathrm{abs}\left(X\left(k\right)\right)$ is the spectrum amplitude, $f_{s}$ is the sampling frequency, and f is the frequency of $x_{n}$. The function of $\mathrm{find}\max\left[k,A\left(k\right)\right]$ is to find the maximum value of $A\left(k\right)$ and return the serial number k corresponding to the maximum value. The calculation process of FFT generally adopts a base 2 butterfly algorithm, and N is set to an integer multiple of 2.

However, because the discrete spectrum is the sampling of the real spectrum, only the amplitude at the discrete points can be observed, and the real maximum value is often not obtained. Therefore, the frequency estimation method based on the FFT has error, and the maximum error is half of the spectral resolution, which is defined by:(5)$$\max\mathrm{ess}=\frac{\delta f}{2}=\frac{f_{s}}{2N}$$
where maxess is the maximum error and δf is the spectral resolution.

Because the frequency range of the LHI signal is 21 MHz, according to Nyquist sampling theorem, $f_{s}$ should be greater than 42 MHz. In order to ensure the quality of the sampling signal, $f_{s}$ is set to 80 MHz. Additionally, the number of sampling points needs to be greater than 2^21^, because the error of frequency estimation needs to be less than 30 Hz. This leads to a serious waste of resources and a long operation time, and so it needs to be optimized.

### 3.2. Principles of COG

On the basis of the discrete spectrum, the frequency can be corrected by using the amplitude of the peak points of the spectrum sequence relative to the neighboring points. Therefore, the accuracy of frequency estimation can be improved by introducing a simple calculation after performing FFT with a smaller number of sampling points. The principle is shown in Figure 3.

By using the monotonicity of the main lobe function near the peak point, the corresponding relationship between the ratio function $f\left(\Delta x-1\right)/f\left(\Delta x\right)$ and Δx is constructed. By calculating the ratio of $y_{k}$ and $y_{k+1}$, Δx can be obtained in reverse. The expression is determined by:(6)$$g\left(x\right)=\frac{f(x)}{f(x+1)},x\in \left[-1,0\right]$$
(7)$$\Delta x=g^{-1}(\frac{y_{k}}{y_{k+1}})$$
(8)$$f_{c}=\left(k-\Delta x\right)\frac{f_{s}}{N}$$
where, f(x) is the main lobe function, $g\left(x\right)$ is the constructed monotonic function, k is the smaller index of discrete spectrum peak and the amplitude of the adjacent point. If $y_{k+1}$ is the discrete spectrum peak, then $f_{c}$ is the amplitude of the adjacent point. $f_{c}$ is the corrected value of $f_{c}$, $f_{c}$ is the corrected frequency. Additionally, when the value of Δx is −0.5, $f_{c}$ is equal to $y_{k+1}$. In this way, the specific expression for f(x) is obtained to enable frequency correction.

### 3.3. Derivation of the Main Lobe Function

The LHI signal contains three beat-notes, and its spectrum is superimposed on the spectrum of three beat-notes, as shown in Figure 4.

By simplifying the LHI signal, the general expression of the superposition signal is obtained:(9)$$x\left(t\right)=\sum_{m=1}^{M}x_{m}\left(t\right)=\sum_{m=1}^{M}A_{m}\cos\left(2\pi f_{m}t+\varphi_{m}\right)$$
where M is the number of signals, and $f_{m}$, $A_{m}$ and $\varphi_{m}$ are the frequency, amplitude, and phase of the mth signal, respectively. $x\left(t\right)$ is expressed as a complex sine:(10)xt=Re∑m=1MAm⋅ej2πfmt+φm

The real signal $x\left(t\right)$ can be obtained by ignoring the imaginary part of the complex function $\hat{x}$. The complex signal $\hat{x}$ is denoted as:(11)$$\hat{x}\left(t\right)=\sum_{m=1}^{M}A_{m}\cdot e^{j\left(2\pi f_{m}t+\varphi_{m}\right)}$$

The sampling truncation of $\hat{x}\left(t\right)$, $f_{s}$ is the sampling frequency, N is the number of sampling points, the sampled signal is represented as:(12)$$\hat{x}\left(k\right)=\sum_{m=1}^{M}A_{m}\cdot e^{j\left(2\pi \frac{f_{m}}{f_{s}}k+\varphi_{m}\right)},k\in \left[0,N-1\right]$$

Denote $\omega_{m}$ as the digital angular frequency. $\omega_{m}$ is determined by:(13)$$\omega_{m}=2\pi \frac{f_{m}}{f_{s}}$$

Substitute Equation (13) into Equation (12) and $x\left(k\right)$ is defined as:(14)$$\hat{x}\left(k\right)=\sum_{m=1}^{M}A_{m}\cdot e^{j\left(\omega_{m}k+\varphi_{m}\right)},k\in \left[0,N-1\right]$$

Typically, a window is used to sample the signal in order to minimize the spectrum leakage caused by data truncation. Signal sampling without an extra window is equivalent to adding a rectangular window. The general expression for the time domain of a window function is:(15)$$w\left(k\right)=\sum_{i=0}^{I}{\left(-1\right)}^{i}\cdot \alpha_{i}\cdot \cos\left(\frac{2\pi i}{N}\cdot k\right),k\in \left[0,N-1\right]$$
(16)$$\sum_{i=0}^{I}\alpha_{i}=1$$
where I is the number of terms in the window function, N is the number of window function points (signal sampling points), and $\alpha_{i}$ is the coefficient of the ith term. For example, $\alpha_{0}=1$, indicates a rectangular window. $\alpha_{0}=0.5,\alpha_{1}=0.5$, indicates a Hann window. $\alpha_{0}=0.42,\alpha_{1}=0.5,\alpha_{2}=0.08$, indicates a Blackman window. $\alpha_{0}=0.35875,\alpha_{1}=0.48829,\alpha_{2}=0.14128,\alpha_{3}=0.01168$, indicate a Blackman–Harris window.

A windowed signal is expressed as:(17)$$x_{N}\left(k\right)=\hat{x}\left(k\right)\cdot w\left(k\right)$$

$x_{N}\left(k\right)$ is the product of $x\left(k\right)$ and $w\left(k\right)$ in the time domain, the discrete Fourier transformation of $x_{N}\left(k\right)$ is convolved in the frequency domain, and the Fourier transformation of $x_{N}\left(k\right)$ is defined as:(18)$$X_{N}\left(e^{j\omega }\right)=F[\hat{x}\left(k\right)\cdot w\left(k\right)]=\frac{1}{2\pi }X\left(e^{j\omega }\right)*W\left(e^{j\omega }\right)=\frac{1}{2\pi }\int_{-\infty }^{\infty }X\left(e^{j\theta }\right)W\left(e^{j\left(\omega -\theta \right)}\right)d\theta$$
where F is the Fourier transform function, and ∗ is the convolution operator. The spectral amplitude of the windowed signal is expressed as:(19)$$\left|X_{N}\left(e^{j\omega }\right)\right|=\left|\sum_{m=1}^{M}A_{m}\cdot W\left(e^{j\left(\omega -\omega_{m}\right)}\right)\cdot e^{j\varphi_{m}}\right|=\sum_{m=1}^{M}A_{m}\cdot \left|W\left(e^{j\left(\omega -\omega_{m}\right)}\right)\right|$$

The above formula ignores the influence of the initial phase of the signal, so this method is susceptible to phase noise. The spectral function of the window is determined by:(20)$$W\left(e^{j\omega }\right)=e^{-j\frac{N}{2}\omega }\left\{a_{0}D\left(\omega \right)+\frac{1}{2}\sum_{i=1}^{I}{\left(-1\right)}^{i}\alpha_{i}\left[D\left(\omega -\frac{2\pi i}{N}\right)+D\left(\omega +\frac{2\pi i}{N}\right)\right]\right\}$$
where $D\left(\omega \right)$ is the Dirichlet kernel, which is defined as:(21)$$D\left(\omega \right)=\frac{\sin\frac{N\omega }{2}}{\sin\frac{\omega }{2}}e^{j\frac{\omega }{2}}$$

The spectrum modulus of $w\left(k\right)$ is denoted as:(22)$$\left|W\left(e^{j\omega }\right)\right|=\alpha_{0}\frac{\sin\frac{N\omega }{2}}{\sin\frac{\omega }{2}}+\frac{1}{2}\sum_{i=1}^{I}\alpha_{i}\left[\frac{\sin\left(\frac{N\omega }{2}-\pi i\right)}{\sin\left(\frac{\omega }{2}-\frac{\pi i}{N}\right)}+\frac{\sin\left(\frac{N\omega }{2}+\pi i\right)}{\sin\left(\frac{\omega }{2}+\frac{\pi i}{N}\right)}\right]$$

Substitute Equation (22) into Equation (19). For the lth frequency component, $\Delta \omega_{il}$ is expressed as the frequency difference between the ith frequency component and the lth frequency component, and the expression for obtaining the lth main lobe function is represented as:(23)$$f_{l}\left(\omega \right)=\sum_{i=1}^{M}A_{i}\cdot \left|W\left(e^{j\left(\omega -\Delta \omega_{il}\right)}\right)\right|=A_{l}\cdot \left|W\left(e^{j\omega }\right)\right|+\sum_{i=1,i\neq l}^{M}A_{i}\cdot \left|W\left(e^{j\left(\omega -\Delta \omega_{il}\right)}\right)\right|$$
(24)$$\Delta \omega_{il}=\omega_{i}-\omega_{l}$$

Let ω=2πx/N, substitute ω into Equation (23), the general formula $f_{m}\left(x\right)$ is determined by:(25)$$f_{m}\left(x\right)=\sum_{r=1}^{R}\left(A_{r}\frac{\sin\left(x\pi +\frac{N\pi \Delta f_{mr}}{f_{s}}\right)}{\sin\left(\frac{x\pi }{N}+\frac{\pi \Delta f_{mr}}{f_{s}}\right)}\left[\sum_{i=0}^{I}{\left(-1\right)}^{i}\frac{\alpha_{i}}{\cos\left(\frac{i\pi }{N}\right)}\right]\right)$$
where R is the number of signal frequencies, $A_{r}$ is the signal amplitude of the rth frequency component, N is the number of sampling points, $\Delta f_{mr}$ is the difference between the mth and the rth frequency component, $f_{s}$ is the sampling frequency, I is the number of terms of the cosine combined window function, and $\alpha_{i}$ is the ith coefficient. Theoretically, Equation (25) is suitable for cosine window functions, and it has the same correction accuracy for different window functions. However, due to the impact of data truncation in the calculation process, the number of terms of the window function does not easily become too large.

## 4. Error Integration

LISA utilizes the direct sequence spread spectrum communication method. It employs the pseudo-random noise code (PRN code) as the communication code to phase modulate the carrier laser. The modulation information is then read out via demodulation, and distance measurement is completed. However, the phase of the signal is unknown during frequency acquisition. The PRN code in the signal acts as a type of phase noise. This creates a deviation between the actual main lobe function and the theoretical function, as shown in Figure 5. Consequently, there will be errors when estimating the actual frequency using the theoretical main lobe function.

In fact, the communication codes used in inter-satellite ranging communication are one or more fixed groups, and the modulation order is fixed. By studying the error of corrected frequency under different sampling starting conditions, it is found that the error of corrected frequency is sinusoidal with the delayed sampling of the signal. Figure 6 shows the relationship between the delayed sampling of the signal and the correction error when the signal is modulated with 2 different groups of communication codes.

By integrating the corrected frequency within the error fluctuation period, the influence of the signal-coupled communication codes or phase noise can be suppressed. The calculation formula is:(26)$$\overset{-}{f_{c}}=\frac{1}{U}\int_{0}^{U}f_{c}\left(u\right)du=\frac{1}{Q}\sum_{q=1}^{Q}f_{c}\left(q\right)$$
where U is the period of error fluctuation, Q is the sampling number within one period, and $\overset{-}{f_{c}}$ is the integral corrected frequency.

## 5. Simulation

The signal shown in Equation (27) is simulated. First, the frequency estimation error of the traditional method is calculated. When the sampling frequency is 80 MHz and the number of sampling points is 65,536, the maximum sampling error is 611 Hz, which is equivalent to half of the spectral resolution. Then, the multi-frequency acquisition method is simulated by using three different window functions, and the acquisition errors of the three methods are similar, which verifies the universality of the proposed method. Finally, the maximum frequency error with and without error integration is compared. The accuracy can be further improved by using error integration, and the probability of the error being less than 1 Hz is about 90 percent.
(27)$$s\left(k\right)=0.9\sin\left(2\pi \frac{f_{m}}{f_{s}}k+0.1m_{prn}\right)+0.05\sin\left(2\pi \frac{f_{u}}{f_{s}}k\right)+0.05\sin\left(2\pi \frac{f_{l}}{f_{s}}k\right)$$
(28)$$f_{m}=2\mathrm{MHz}+\Delta f\times \left(M-1\right)$$
where $f_{m},f_{u},f_{l}$ are the frequency of the three beat-notes, $f_{u}=f_{m}+1\mathrm{MHz}$, $f_{l}=f_{m}-1\mathrm{MHz}$, $f_{s}$ = 80 MHz, N = 65,536, $m_{prn}$ is communication codes. The method was tested $f_{m}$ from 2 MHz to 19,997,989.4 Hz. Additionally, the number of test points M = 3383. Δf = 5321.7 Hz. Table 1 shows the maximum error of the frequency acquisition simulation.

The simulation results without error integration are shown in Figure 7. The acquisition errors of three kinds of window function are similar, the maximum acquisition error of the carrier beat-note is about 2.4 Hz, and the maximum acquisition error of the two side beat-notes is about 36 Hz. Compared with 610 Hz of the FFT method, the precision of two side beat-notes is improved by an order of magnitude and the carrier beat-note is improved by two orders of magnitude. This method is only compared with the FFT method because it is an improvement on FFT. Of course, this method is not the most accurate method, but it is suitable for space gravitational wave detection. This is because this method requires less computation, so it is suitable for real-time processing.

The simulation results of error integration are shown in Figure 8. The acquisition errors under the three window functions are similar. The maximum acquisition error of the carrier beat-note is below 0.4 Hz, and the maximum acquisition error of the two side beat-notes are below 4.6 Hz. Compared with the simulation results without error integration, the acquisition accuracy is increased by about an order of magnitude. Additionally, the probability of two side beat-notes between −1 Hz and 0 is about 0.9. Figure 9 shows the probability distribution of frequency acquisition errors, where P represents the probability.

## 6. Implementation of the Algorithm

The multi-frequency acquisition algorithm consists of clock module, communication module, signal windowing sampling (SWS) module, discrete spectral amplitude calculation (DSAC) module, peak search (PS) module, frequency correction (FC) module, and error integration (EI) module, as shown in Figure 10.

The clock module divides the 20 MHz external clock into the 80 MHz system clock and 320 MHz AD sampling clock; the communication module uses an RS 232 serial port to read the operation instructions sent by the PC and receive the calculation results for the three frequencies; the SWS module adopts the ADI’s high-speed ADC chip, with a sampling rate of 80 MHz and a 16-bit digital signals output, and a serial data rate of 320 Mbps. The window is Hanning, and the sequence of 16-bit window functions is stored in ROM. Windowing is realized by multiplication, and the 32-bit windowed signal is the output. The DSAC module uses the existing algorithm packages: the FFT algorithm and CORDIC algorithm calculate the modulus of the FFT results; the PS module calculates the peak value and peak index of the spectrum by peak search algorithm, and the flow chart of the algorithm is shown in Figure 11. The FC module stores the frequency correction curve in ROM, and calculates the inverse solution of the amplitude ratio using a division algorithm. Finally, the EI module further reduces the acquisition error based on Equation (26).

## 7. Experimental Facilities and Test Results

The experimental facilities of the frequency acquisition experiment included an LHI signal simulation system for space gravitational wave detection, a frequency acquisition algorithm experimental circuit board, an external reference clock, a power supply, and a PC, as shown in Figure 12.

The signal simulation system was used to generate the simulated LHI signal. The signal is shown in Equation (27). $f_{m}$ ranged from 2 MHz to 20 MHz. The number of test frequency M was 34. The frequency change interval Δf was 532,170 Hz. Each frequency was acquired 10 times. Figure 13 shows the results of the frequency acquisition experiment. The maximum error of the carrier beat-note was less than 1 Hz. Additionally, the maximum error of two side beat-notes was less than 10 Hz. The acquisition time was 125 ms. Comparing Figure 8 and Figure 13, the experimental acquisition error was slightly larger than the theoretical simulation value. This was caused by the experimental circuit and the truncation error of AD sampling, and further analysis of this influence is needed in the future. At present, the maximum acquisition error meets the task requirement of less than 30 Hz. However, in the future, with the improvement of index requirements, higher requirements may be put forward for acquisition accuracy.

## 8. Conclusions

In this paper, a multi-frequency acquisition metod of LHI signals is presented, and a system platform was built. A signal simulation system was used to output an LHI simulation signal, which was analyzed using the frequency acquisition method in FPGA, and then the frequency was obtained using a PC. In addition, the simulation results of frequency acquisition under different window functions show that the frequency acquisition errors are basically the same, which verifies the universal applicability of this method. The experimental results show that the method can reduce the fluctuation of communication codes, and acquire signals from 1 MHz to 21 MHz. Compared with FFT, the maximum acquisition error of this method is reduced to 10 Hz, which meets the requirements of current frequency acquisition tasks. However, due to the noise of the experimental circuit or the truncation error of AD sampling, the acquisition error of the experimental results was slightly larger than that of the theoretical simulation. Therefore, in future work, it is necessary to analyze the influence of noise and clarify the influence mechanism, so as to apply the method better. In addition, we need to further optimize the structure and computational complexity of the method, so as to improve the computational efficiency in engineering while maintaining accuracy. Finally, it is worth noting that the method is not only suitable for space gravitational wave detection, but also for other occasions of high-accuracy acquisition of multi-frequency signals, such as harmonic detection of power signals.

## Figures and Tables

**Figure 1 sensors-23-03124-f001:**
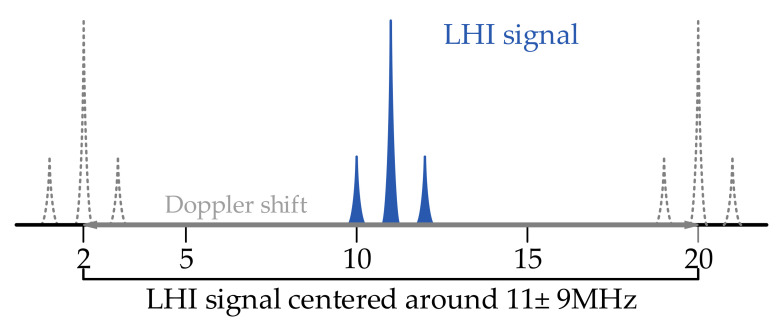
Illustration of the effect of Doppler shift. The LHI signal consists of three beat-notes. The initial setting of the frequency of the carrier beat-note is 11 MHz and the frequencies of side beat-notes are 11 MHz ± 1 MHz. Affected by Doppler shift, three frequencies are unknown. The frequency of carrier beat-notes ranges from 2 MHz to 20 MHz, and the range of frequency variation of LHI signal is 1 MHz to 21 MHz.

**Figure 2 sensors-23-03124-f002:**
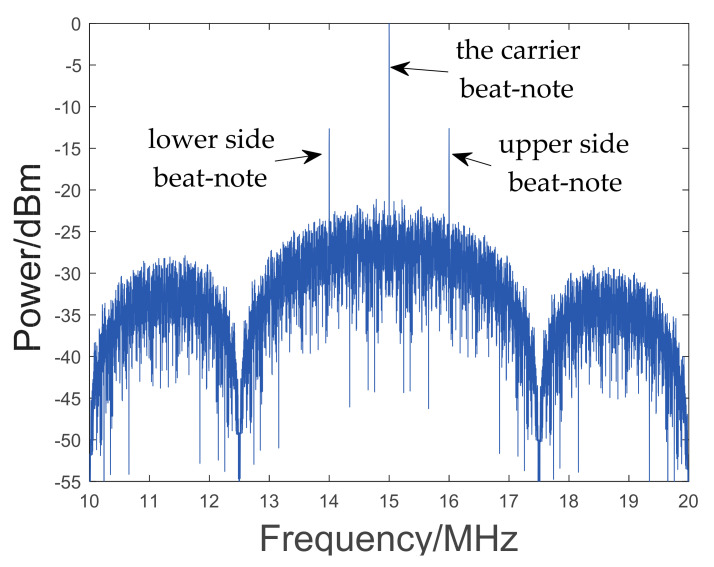
The logarithmic spectrum of the LHI signal. The frequency of carrier beat-note is 15 MHz, including the Doppler shift of 4 MHz. Because the signal modulates the communication code of 2.5 MHz, the spectral envelope is symmetrical about carrier beat-note and the distance is 2.5 MHz.

**Figure 3 sensors-23-03124-f003:**
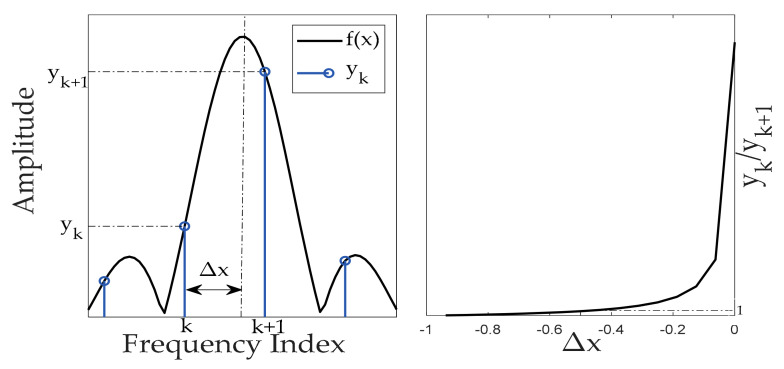
Principle of spectrum correction. $f\left(x\right)$ is the main lobe function, k is the smaller index of discrete spectrum peak and the amplitude of the adjacent point. $y_{k+1}$ is the peak of the spectrum, $y_{k}$ is the amplitude of the neighboring point, Δx is the corrected value of k. Construct the monotonic function $y_{k}/y_{k+1}$ to calculate Δx.

**Figure 4 sensors-23-03124-f004:**
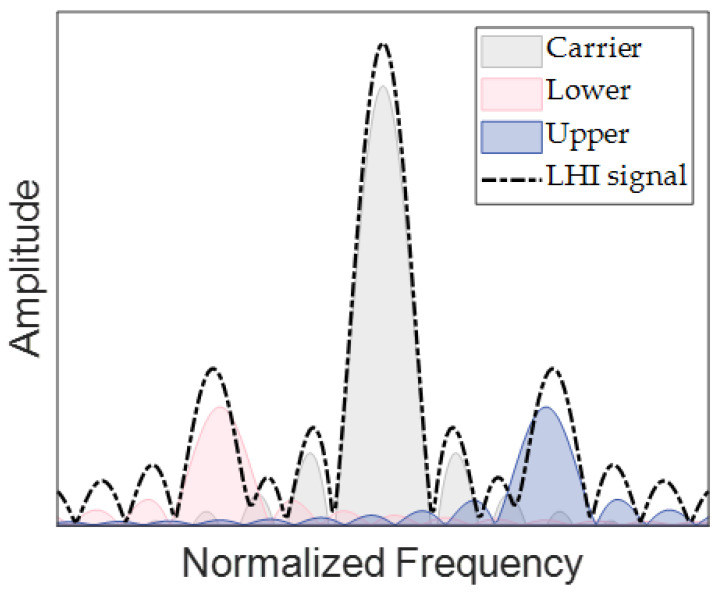
Illustration of spectrum superposition. Gray represents the carrier beat-note, pink represents the lower side beat-note, blue represents the upper side beat-note, and the black dashed line represents the LHI signal.

**Figure 5 sensors-23-03124-f005:**
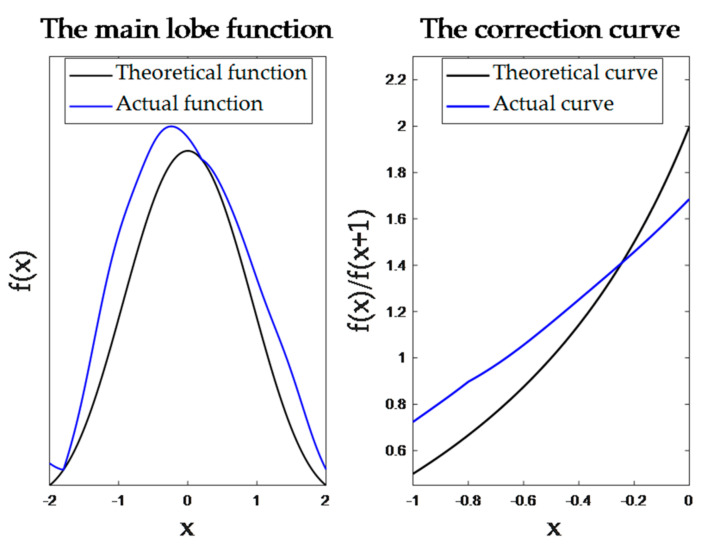
The effect of communication codes on the method. x is the corrected value of peak index.

**Figure 6 sensors-23-03124-f006:**
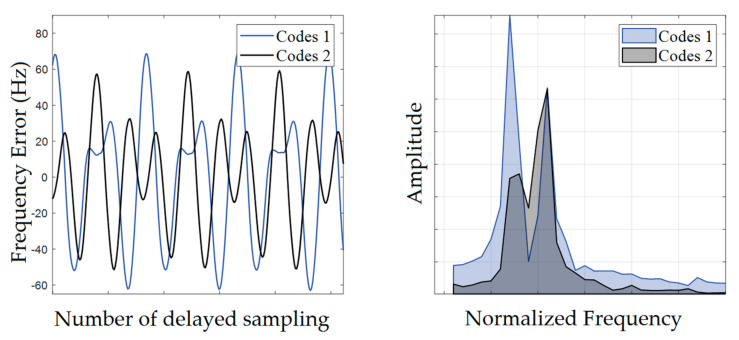
To the left is the relationship between signal delay sampling and frequency errors. Let the signal sequence be $x\left(k\right)$, the number of points used in each calculation be N, and the signal sequence for the ith calculation be $x\left(i\right)$ to $x\left(i+N-1\right)$. To the right is the spectrum of the frequency error. By analyzing the composition of the spectrum, it is possible to calculate the error fluctuation period.

**Figure 7 sensors-23-03124-f007:**
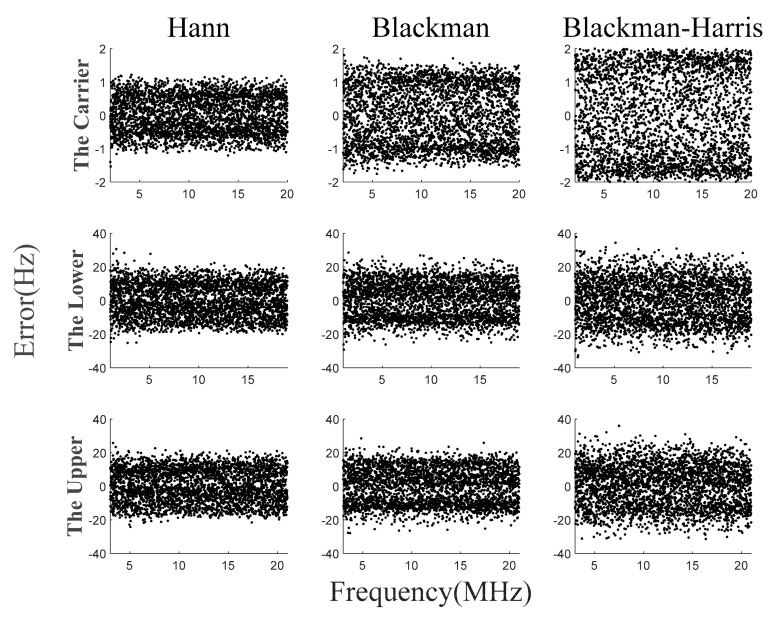
Simulation results of frequency acquisition algorithm without error integration.

**Figure 8 sensors-23-03124-f008:**
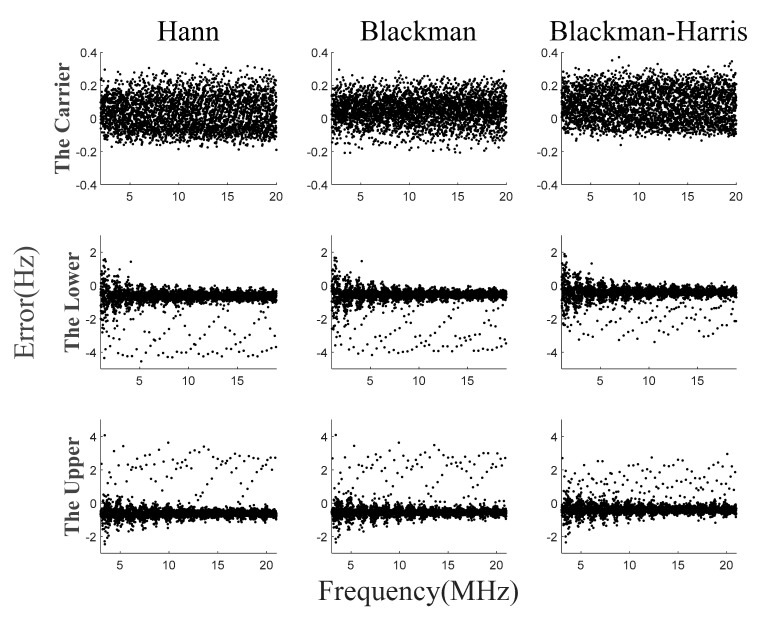
Simulation results of frequency acquisition algorithm with error integration.

**Figure 9 sensors-23-03124-f009:**
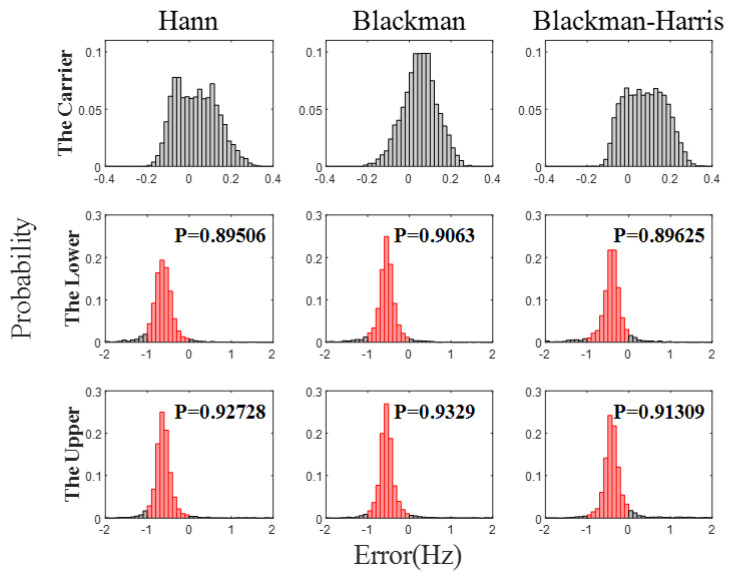
The probability distribution of the frequency acquisition error. Red represents the probability that the error is between −1 Hz and 0 Hz.

**Figure 10 sensors-23-03124-f010:**
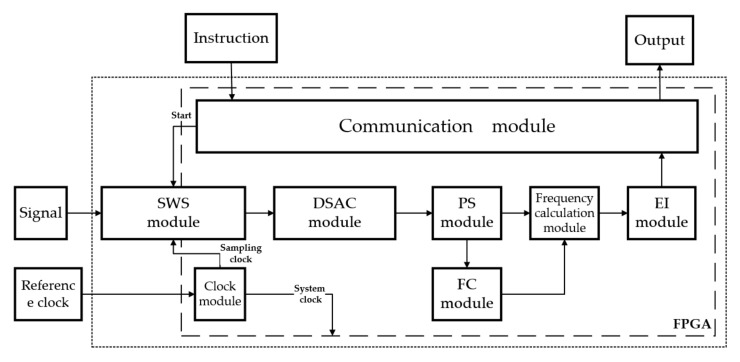
Block diagram of the algorithm.

**Figure 11 sensors-23-03124-f011:**
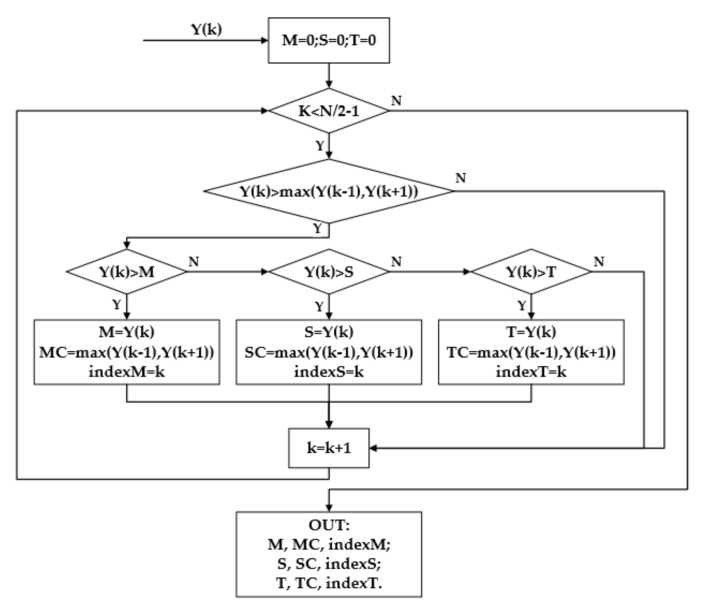
The peak search algorithm logic. M, S and T are the maximum, second and third peak of the amplitude spectrum, MC, SC and TC are the corrected amplitudes, and index is the corresponding spectral sequences.

**Figure 12 sensors-23-03124-f012:**
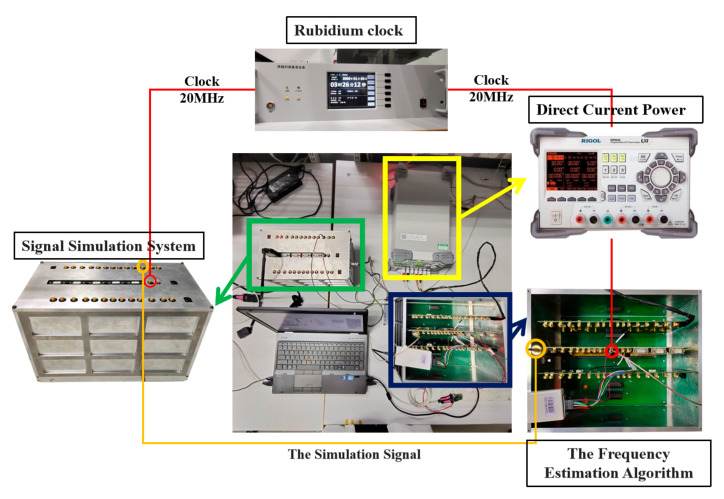
The system platform.

**Figure 13 sensors-23-03124-f013:**
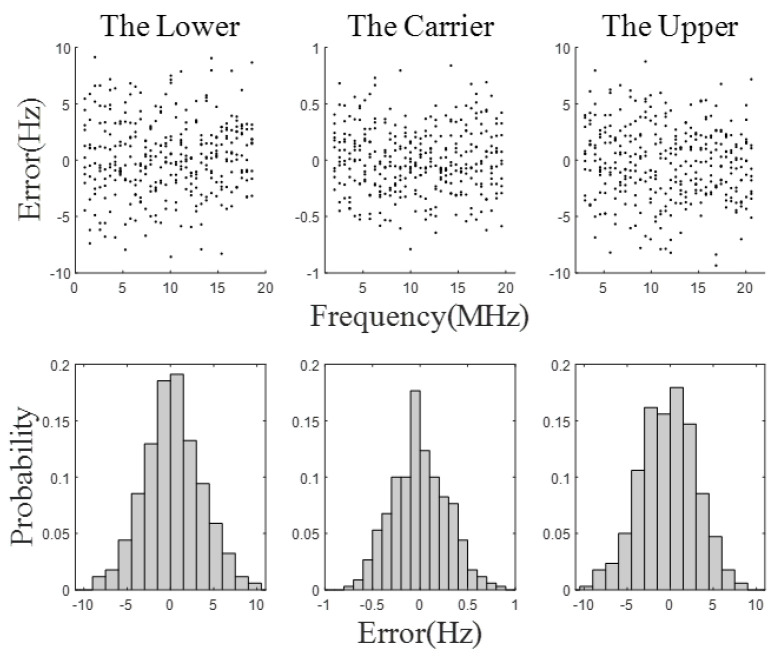
The experimental results.

**Table 1 sensors-23-03124-t001:** The maximum error of the frequency acquisition simulation.

Method		Error of f_m_	Error of f_u_	Error of f_l_
FFT		610.5531 Hz	610.5632 Hz	610.7658 Hz
frequency correction	Hann	1.5336 Hz	25.7858 Hz	30.6588 Hz
	Blackman	1.8053 Hz	28.5197 Hz	43.9575 Hz
	Blackman–Harris	2.3936 Hz	35.9808 Hz	37.8291 Hz
error integration	Hann	0.3325 Hz	4.0745 Hz	4.5455 Hz
	Blackman	0.2945 Hz	4.0925 Hz	4.1662 Hz
	Blackman–Harris	0.3708 Hz	2.9534 Hz	3.3880 Hz

## Data Availability

The data and the source code are publicly available on https://github.com/zhenpengwang/data_Frequency_Estimation (accessed on 30 December 2022).

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
