# Peer review of "Beat-Notes Acquisition of Laser Heterodyne Interference Signal for Space Gravitational Wave Detection"

_sensors, 2023, doi:10.3390/s23063124_

Round 1
Reviewer 1 Report (Previous Reviewer 1)
The writing of the manuscript is improved substantially. Problem Motivation should be improved to show readers that why this problem is important compared to the existing work. Currently, this method is shown to be better only compared to the FFT method. In case there is no other closely relevant literature available then the authors should clearly justify this.
As a result section, results are being shown with a high improvement compared to FFT but authors should discuss the improvement in the context of the theory section. For example discuss, why 10-time improvement is achieved in Table 1 and then Figure 7/8. Should we suppose that only the FFT method is available for comparison?
Author Response
Please see the attachment.

Reviewer 2 Report (New Reviewer)
In the manuscript titled "Beat-notes Acquisition of Laser Heterodyne Interference Signal for Space Gravitational Wave Detection", the authors introduce, analyze, and implement a method to estimate frequencies in the context of laser heterodyne interference (LHI) signals for space-based gravitational wave detectors. The proposed center of gravity estimation method overcomes previous limitations from traditional FFT-based estimation, specifically in the overall accuracy requirement for space-based gravitational wave detectors suffering from frequency shifts such as Doppler shifts.
General comments:
The authors may want to consider expanding the discussion at the end regarding the experimental demonstration and its mismatch between the extensive numerical simulations.
Specific comments:
- In Section 3, subsection 3.1, around line number 126-127 a sentence states "the error of frequency estimation needs to be less than 30 Hz" to motivate the proposed COG method, but the broader context is not immediately apparent. The high-accuracy requirement in frequency estimation is also mentioned as early as lines 54-55, but this is the first time a number figure is quoted-- it would therefore be desirable to place the readers into context to have this figures of merit appear before, or be given a better context. This is a critical issue.
- Section 4 provides an example of error sources using communication codes. Other than a qualitative description, this section seemed brief and the authors may consider merging with Section 5 (simulations). This is not a critical issue.
- In Figure 15 showing experimental results, the authors may want to consider expanding the caption to highlight the qualitative aspects of the result. Alternatively, and in relation to my general comment, the authors may simply want to expand their discussion to match the extent of analytical and numerical counterpart sections (Sections 3 and 5). This is a critical issue.
The manuscript needs some minor editing in order to improve, so I would like to recommend this manuscript for publication provided the authors address the two specific comments above marked as critical.
Author Response
Please see the attachment.

Reviewer 3 Report (New Reviewer)
See attached PDF file.

Round 2
Reviewer 3 Report (New Reviewer)
The authors have changed the manuscipt taking into account all comments. So, the paper has now my recommendation for publication.
This manuscript is a resubmission of an earlier submission. The following is a list of the peer review reports and author responses from that submission.
Round 1
Reviewer 1 Report
In this paper, the authors claim to propose a frequency estimation method for the laser heterodyne interference signal and then they claim that they have built a system platform to test the proposed algorithm in real-time. However, the authors have completely failed to present the problem and aim of the paper, therefore, it is not clear what they have done to be claimed as a new method. In some places, they mention that they have improved the estimation of frequency by using Fast Fourier Transform (FFT) instead of Discrete Time Fourier (DFT) Transform. But this claim is not clear from the over all writing. Then they also claim that they have also done real-time experiments to test their proposed algorithm.
My comments:
The authors have failed to present what is the main problem in the relevant current state of the art, and then what is the exact contribution of this paper. Missing this, the readers may fail to judge the novelty of this work.
English writing and presentation of the paper are also very bad. This aspect further makes it very hard for the reader to grasp the main idea due to bad grammar, redundancies, etc. In my opinion, with such bad writing, the paper should not have been sent to the reviewer.
Therefore, the grammar needs to be corrected. For example, just focus on the Abstract and Introduction where sentences are long, no connection between the sentences, no flow in writing, and bad grammar. The authors must understand the attributes of the Abstract and Introduction via basic principles of technical writing.
The authors claim in some places that they have developed a system with the help of FFT compared to DFT and improved performance. In the methodology parts of the paper, the proposed method is not clearly presented. For example in Figure 3, page 4, they have rather presented a system with DFT. The authors are strongly advised to rewrite their own method with more clarity and to-the-point discussion.
In terms of computational complexity, although, FFT is better DFT, but authors should explain why FFT performs better than DFT in terms of better estimation.
The results are not clear. For instance, the two Simulation figures (Fig 10 and Fig 11) are not clearly explained and the role of FFT is missing in these two figures. As it is mentioned earlier part of the paper that the comparison of three other methods was made between FFT. In figure three it is mentioned that the “Blackman-Harris window has a good effect” but it has the highest error, So how is the highest error a good one?
It is mentioned that the accuracy of “Simulation of error integration” is 10 times higher than without integration. But how is that accuracy calculated? No information about the calculation is given. Similarly, how is the probability and probability error calculated? No such information is given. The performance of estimation methods is compared and analyzed with mean squared errors rather than plotting a histogram of Errors.
No comparison with DFT, Learning based frequency estimation is made as it is mentioned in the paper that we are comparing our technique with these techniques. In fact, no comparison is made with any technique and in the simulation result section, just the results derived from their own paper are mentioned. So no comparison with the state of the art.
Just like the Abstract and Introduction, the authors must check what is the purpose of the Conclusion Section and what should be written.
References are not properly formatted and papers should be cited from recent literature published in high-rated Journals.
Reviewer 2 Report
The author proposed a frequency estimation algorithm based on multiple cosine combined windows. The maximum frequency estimation error of beat notes calculated by this method was reduced from 610 Hz of the discrete Fourier transform (DFT) algorithm to 10 Hz. The Paper is clear, innovative, relevant, and can be published. However, there are some inadequacies in the description, so I propose revising it according to the comments below.
1) As far as I know, the classical windows commonly used in the FFT process to reduce spectrum leakage are the Rectangular window, Triangular window, Hanning window, Gaussian window, Hamming window, and Blackman window. In the simulations in Section 5, why did the authors choose Hanning, Blackman, and Blackman-Harris windows to analyze errors?
In other words, in the experiments in Section 6, the windowing module of the FPGA platform seems to refer to the function selection in Section 5 to achieve better errors, that is, is there a better performing window function before applying the frequency estimation algorithm?
2) Line 240, "The simulation of error integral are shown in Figure 11." But the following sentence is the description of Table 2. I recommend the author adjust the order of sentences to increase the logic and readability of the article.
3) In line 268, the author mentioned that the amplitude spectrum calculation module uses the CORDIC algorithm, so what about the FFT module in line 267? Is the Radix-2 Butterfly algorithm? I recommend the author make an addition to improve consistency.
4) Please check for formatting errors like “Mhz” in line 101 and line 259.
Reviewer 3 Report
The paper describes an analysis of a multi-cosine windowing technique to improve frequency estimation in heterodyne signals for spacecraft ranging. While the problem discussed in this paper for sure deserves attention, I think that the novel aspect of the presented technique, i.e., the multi-cosine windowing technique, is insufficient for a publication. Furthermore, the paper is hard to read due to grammar errors and inaccurate physics terminology. In conclusion, primarily because I do not see a major advance in this paper, I do not recommend this paper for publication in Sensors.